PERSPECTIVE

# A systems biology approach towards understanding and treating non-neovascular age-related macular degeneration

James T. Handa[1], Cathy Bowes Rickman[2], Andrew D. Dick[3,4], Michael B. Gorin [5,6], Joan W. Miller [7], Cynthia A. Toth[2], Marius Ueffing[8], Marco Zarbin[9] & Lindsay A. Farrer[10]

Age-related macular degeneration (AMD) is the most common cause of blindness among the elderly in the developed world. While treatment is effective for the neovascular or "wet" form of AMD, no therapy is successful for the non-neovascular or "dry" form. Here we discuss the current knowledge on dry AMD pathobiology and propose future research directions that would expedite the development of new treatments. In our view, these should emphasize system biology approaches that integrate omic, pharmacological, and clinical data into mathematical models that can predict disease onset and progression, identify biomarkers, establish disease causing mechanisms, and monitor response to therapy.

Age-related macular degeneration (AMD) is the world's leading cause of blindness among the elderly[1]. It is projected that the number of people with AMD worldwide will be 196 million in 2020, increasing to 288 million in 2040, with the pooled prevalence of early, late, and any AMD to be 8.01% (95% CI 3.98–15.49), 0.37% (0.18–0.77), and 8.69% (4.26–17.40), respectively[1]. While AMD is currently more prevalent in Europe and North America than Asia, given that Asia accounts for more than 60% of the world's population, the largest projected number of AMD cases will occur in Asia[1]. In the US alone, approximately 11 million people have AMD, a prevalence that is similar to that of all invasive cancers combined, and more than double of that of Alzheimer's disease[2].

AMD is classified into two forms, a non-neovascular or "dry" form and a neovascular or "wet" form (Fig. 1). In the wet form, rapid, severe vision loss can occur due to the development of new blood vessels from the choroid into the subretinal space, within Bruch's membrane, or in the subretinal pigmented epithelial (RPE) space that can leak fluid, hemorrhage, and with time, develop fibrosis around these neovascular tufts. In dry AMD, vision loss is typically gradual. Dry

[1] Wilmer Eye Institute, Johns Hopkins University, Baltimore 21287 MD, USA. [2] Department of Ophthalmology, Duke University Medical Center, Durham 27708 NC, USA. [3] Translational Health Sciences (Ophthalmology), University of Bristol, Bristol BS8 1TH, UK. [4] University College London, Institute of Ophthalmology and the National Institute for Health Research Biomedical Research Centre, Moorfields Eye Hospital and UCL-Institute of Ophthalmology, London WC1E 6BT, UK. [5] Department of Ophthalmology, Jules Stein Eye Institute, David Geffen School of Medicine, UCLA, Los Angeles 90095 CA, USA. [6] Brain Research Institute, UCLA, Los Angeles 90095 CA, USA. [7] Retina Service, Massachusetts Eye and Ear, Harvard Ophthalmology AMD Center of Excellence, Department of Ophthalmology, Harvard Medical School, Boston 02114 MA, USA. [8] Department of Ophthalmology, Institute for Ophthalmic Research, University of Tübingen, Tübingen D-72076, Germany. [9] Institute of Ophthalmology and Visual Science, New Jersey Medical School, Rutgers University, Newark 07103 NJ, USA. [10] Departments of Medicine (Biomedical Genetics), Neurology, Ophthalmology, Epidemiology, and Biostatistics, Boston University Schools of Medicine and Public Health, Boston 02118 MA, USA. [11] These authors jointly supervised this work: James T. Handa, Lindsay A. Farrer. Correspondence and requests for materials should be addressed to J.T.H. (email: jthanda@jhmi.edu) or to L.A.F. (email: farrer@bu.edu)

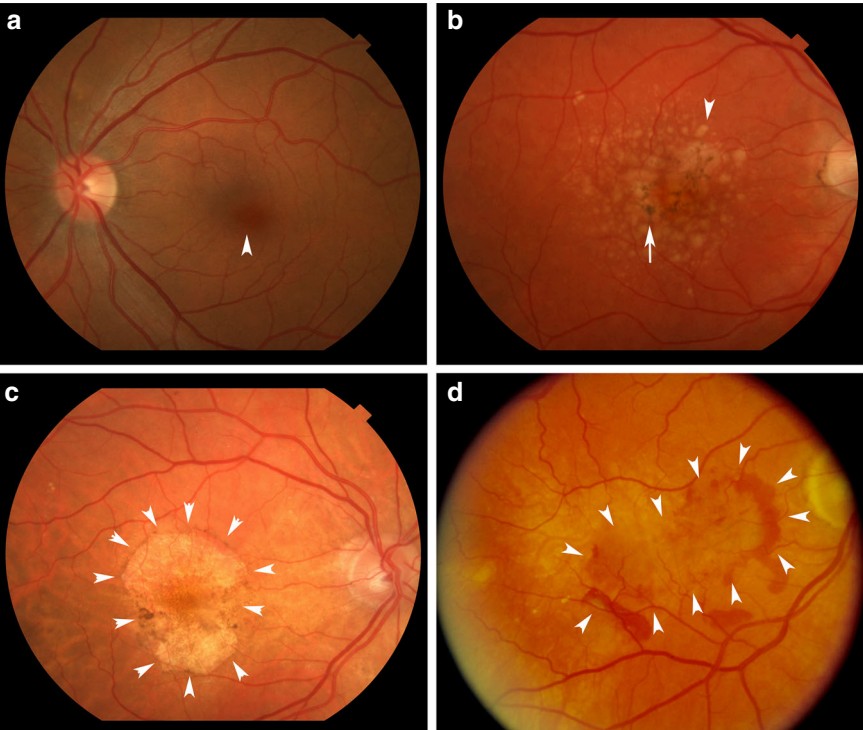

**Fig. 1** Spectrum of AMD. **a** Fundus photograph of a normal left macula. Arrowhead points to the fovea. **b** Right eye with intermediate, dry AMD. Arrowhead points to a typical large druse while arrow points to RPE hyperpigmentation. **c** Right eye with geographic atrophy. Arrowheads outline the area of GA. **d** Right eye with neovascular AMD. Arrowheads outline the area of choroidal neovascularization that is partially outlined by subretinal hemorrhage

AMD is defined clinically by the presence of at least intermediate-size yellow sub-RPE deposits called drusen (63 μm or larger in diameter), RPE pigmentary abnormalities, and subretinal deposits called reticular pseudodrusen[3]. These pigmentary abnormalities are the clinical manifestation of RPE degeneration, which can ultimately culminate in death of the RPE and of the overlying photoreceptors. Multiple medium-sized drusen, large-sized drusen, RPE pigmentary changes, and AMD duration are independent risk factors for developing late AMD[3]. In late, dry AMD or geographic atrophy (GA), patches of RPE cell loss become confluent. When GA involves the fovea, vision loss is severe.

While successful treatment using anti-vascular endothelial cell growth factor treatment is available for wet AMD[4], no effective prevention or treatment is available for dry AMD. The Age-related Eye Disease Study (AREDS) trials demonstrated that antioxidant micronutrient supplements given to intermediate AMD patients modestly reduced the risk of developing advanced disease, and in particular, wet AMD[5,6]. Recently, lampalizumab, an inhibitor against complement Factor D, showed some efficacy in slowing GA progression in a Phase 2 study[7], but this result was not confirmed in a Phase 3 study. A Phase 2 study conducted by Apellis using an inhibitor of complement factor 3 activation (APL-2) demonstrated a reduction in the rate of GA progression but was also associated with an increased risk of wet AMD in a subset of patients[8]. A number of other approaches for treating GA have failed in human trials (e.g., visual cycle inhibitors, emixustat, Acucela, sustained release of neurotrophic factors, NT501, Neurotech, and complement pathway inhibitors, eculizumab, Soliris)[7,9].

The lack of preventive measures and treatment for dry AMD underscores the importance of gaining a better understanding of its pathobiology. Prior research has implicated strong roles for inflammation, and complement in particular, mitochondrial dysfunction, oxidative stress, lipid abnormalities, and cell death in dry AMD pathobiology, but their precise mechanisms are unclear. Furthermore, the relative magnitude and temporal contributions of these factors remain elusive. Due to the multifactorial etiology, effective management of dry AMD may require multiple targets that differ for prevention and therapy of early, intermediate, and late stage disease. These targets will likely result from the elucidation of the mechanistic pathways that are critically involved at each disease stage. The National Advisory Eye Council established a working group to evaluate the current knowledge on dry AMD pathobiology and propose future research directions that would expedite the development of new treatments and the purpose of this perspective is to report on the findings from this working group. The intention is to raise awareness of the impact of AMD on public health, review the current understanding of the pathobiology of this disease, offer future research directions that focus on unbiased systems approaches, encourage the continued efforts of dedicated vision scientists who focus on dry AMD, and encourage the broader scientific community to join in a collaborative effort to develop therapies for this complex and debilitating disease.

## Hallmark pathological changes in dry AMD

The macula is the central area of the retina, 6 mm in diameter, which contains more than one layer of ganglion cell nuclei (Fig. 2). The central macula or fovea, 0.8 mm in diameter, is cone-dominated while the surrounding parafovea is rod-dominated. In the normal aging parafovea, the RPE become enlarged and often multinucleated[10]. In early, dry AMD, parafoveal rods die before either the RPE or cones[11]. With advancing dry AMD, the RPE degenerates with severe changes in cell shape, often becoming multilayered, and dissociated from Bruch's membrane with migration into the retina or below Bruch's membrane[12]. These changes suggest that RPE dysfunction has a central role in the development of photoreceptor loss in dry AMD.

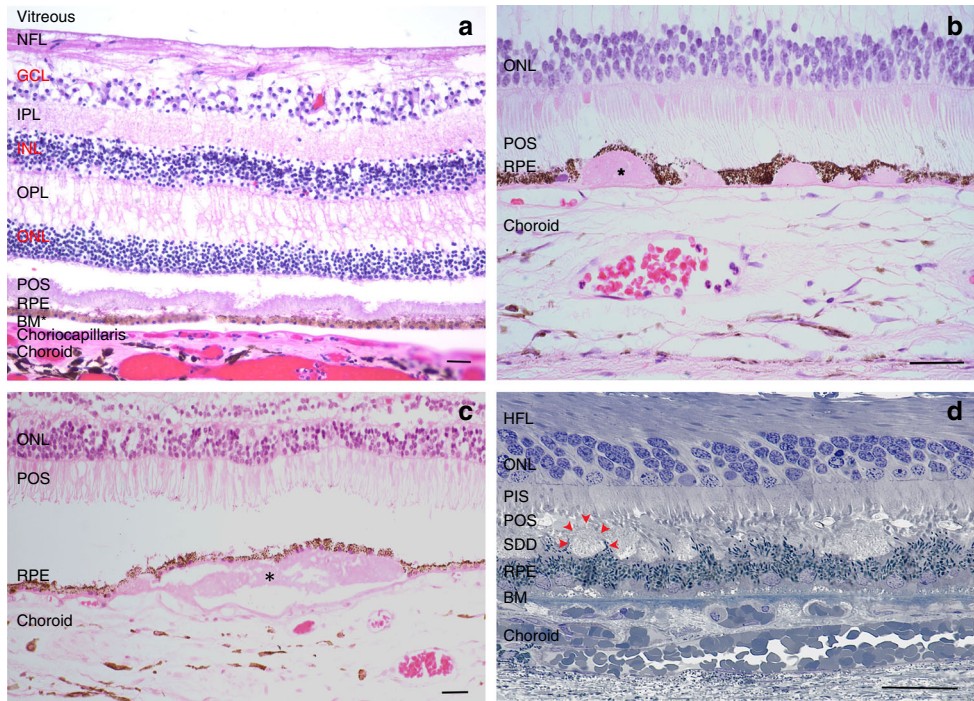

**Fig. 2** Histological cross-section of a human macula. **a** Normal macula. **b** Hard drusen (*) within Bruch's membrane. **c** A soft druse (*) within Bruch's membrane. (**b** and **c** are courtesy of Philip Luthert, MBBS, UCL Institute of Ophthalmology.) **d** Subretinal drusenoid deposits (SDDs) between shortened photoreceptor outer segments and the RPE. Arrows highlight one of several SDDs (Courtesy of Christine Curcio, Ph.D., University of Alabama Birmingham and Project MACULA AMD Histopathology resource (http://projectmacula.cis.uab.edu/). HFL Henle fiber layer, NFL nerve fiber layer, GCL ganglion cell layer, IPL inner plexiform layer, INL inner nuclear layer, OPL outer plexiform layer, ONL outer nuclear layer, POS photoreceptor outer segment, RPE retinal pigment epithelium, BM Bruch's membrane. Bar = 25 μm

Coincident with RPE changes, Bruch's membrane develops basal deposits or accumulations of heterogeneous debris[12]. With aging, the inner Bruch's membrane accumulates apolipoprotein B100-containing lipoproteins[13], which stimulate inflammatory infiltration, accumulation of cellular debris, and the formation of basal deposits. Basal laminar deposits accumulate between the RPE and its basement membrane, and are associated with dry AMD when they become thick and composed of heterogeneous debris. Basal linear deposits, which form within Bruch's membrane's inner collagenous layer, are specific to dry AMD. Nodular-shaped basal linear deposits and focal basal laminar deposits accumulations are visualized as soft drusen on clinical exam.

The choriocapillaris is the sinusoidal capillary network of the choroidal circulation that is adjacent to Bruch's membrane. Choriocapillaris endothelium has fenestrations that enable the bidirectional movement of fluid and macromolecules with the RPE and outer retina. Early dry AMD is characterized by loss of the choriocapillaris, which provides oxygen and nutrients for RPE and photoreceptor survival. Choriocapillaris loss precedes RPE atrophy and correlates with drusen size and density, thus implicating choriocapillaris dysfunction in RPE survival and drusenogenesis[14]. Finally, reticular pseudodrusen are extracellular deposits that accumulate in the subretinal space. While initially overlooked, improved imaging has shown that reticular pseudodrusen are part of a continuum of pathology that is referred to as subretinal drusenoid deposits (SDDs). The presence and progression of SDD can predict disease advancement to GA[15].

**Stressors and pathways implicated in dry AMD pathobiology**
**Oxidative stress.** Photoreceptors and the RPE have high metabolic activity that results in reactive oxygen species (ROS) generation. The high metabolic demand requires a high oxygen partial pressure of 70–90 mmHg[16], and the unique, photo-oxidative stress from light exposure make the macula a high oxidative stress microenvironment. In addition, several lifestyle choices such as cigarette smoking, and high fat or high glycemic index diets, add to the oxidative stress burden, and are associated with AMD risk[17–19]. The AREDS trials showed that, among patients with intermediate AMD, antioxidants lower the risk of developing advanced AMD[5,6]. Finally, genetic variants in oxidative stress-related genes, including *MTND2*LHON-4917G, NADH subunits, *SOD2*, and *PPARGC1A*, are associated with AMD risk[20–22].

**Lipids.** Because lipids can occupy more than 40% of drusen volume[23], and polymorphisms in several lipid-related genes, including *LIPC, CETP, ABCA1,* and *APOE*, are associated with AMD risk[24–27], lipids play a critical role in drusenogenesis. The RPE accumulates cholesterol either from phagocytosis of photoreceptor outer segments or from the ingestion of lipoproteins from the circulation. The RPE recycles cholesterol back to the photoreceptor or eliminates it through reverse cholesterol transport by effluxing cholesterol to ApoAI-I to form a high-density lipoprotein (HDL)[28]. With lipoproteinemia, lipoprotein ingestion is excessive, and the RPE becomes cholesterol-overloaded. Should the reverse cholesterol transport fail, the RPE will secrete apoB100 lipoproteins into Bruch's membrane[13,29]. With aging, Bruch's membrane accumulates advanced glycation endproducts, which induces lipoprotein lipase, causing the retention and oxidation of lipoproteins[30,31]. Hydroxyapatite surrounds the oxidized lipoproteins, which get coated with lipids and inflammatory proteins to promote drusen growth[32]. Since cones contain more cholesterol than rods, drusen tend to accumulate in the cone-rich fovea[33].

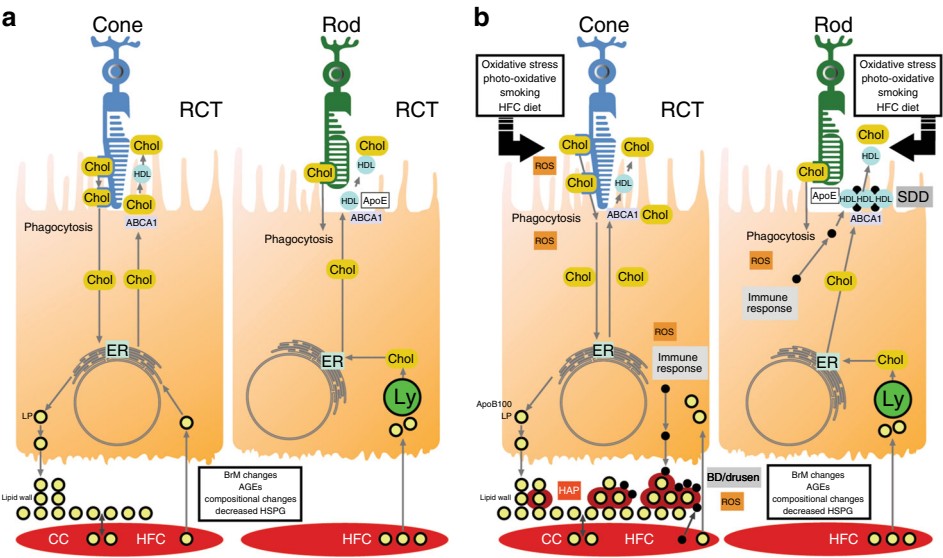

**Fig. 3** Schema of the role of lipids in drusen and subretinal drusenoid deposit formation. **a** In the cone-rich fovea, the high cholesterol content of cones, along with lipids derived from the circulation due to a high cholesterol diet, induces apoB100 lipoprotein formation that is basally secreted. Due to age-related Bruch's membrane changes, lipoproteins accumulate, forming the lipid wall. In the rod-rich parafovea, reverse cholesterol transport mediated through ABCA1 releases cholesterol to apoE and apoA1, forming high-density lipoproteins for the recycling of lipids including docohexanoic acid and cholesterol to rods. **b** Hydroxyapatite forms around retained lipids and lipoproteins, and combined with lipoprotein oxidation, induces an inflammatory response with the accumulation of inflammatory debris, leading to basal deposit and drusen formation. In the subretinal space, dysfunction of reverse cholesterol transport can lead to lipid accumulation, which induces an inflammatory response, forming subretinal drusenoid deposits. AGEs advanced glycation endproducts, BD basal deposit, BrM Bruch's membrane, CC choriocapillaris, CEP carboxyethyllysine, Chol cholesterol, ER endoplasmic reticulum, HAP hydroxyapatite, HDL high-density lipoprotein, HFC high-fat cholesterol diet, HSPG heparan sulfate proteoglycan, Lp apoB100 containing lipoprotein, Ly lysosome, Mono/macro monocyte/macrophage, RCT reverse cholesterol transport, ROS reactive oxygen species, SDD subretinal drusenoid deposit; yellow circle surrounded by red, apoB100 containing lipoprotein with surrounding HAP

SDDs contain unesterified cholesterol and apoE, which suggests that HDLs are involved in their formation[34,35]. Photoreceptors and the RPE have an active bidirectional cholesterol transport where HDLs accumulate cholesterol released into the subretinal space and cycles cholesterol between the RPE and photoreceptors[28]. With RPE dysfunction, cholesterol-laden HDLs accumulate in the subretinal space, triggering inflammation, and the accumulation of complement factors, vitronectin, and immune cells[36]. Figure 3 summarizes the role of lipids in AMD lesion formation.

**Inflammation and innate immunity**. Altered immune responses that lead to destructive neuroinflammation are thought to contribute to the dry AMD phenotype. Parainflammation is a low-grade cytoprotective adaptation to local stress that is intermediate between immune-mediated homeostasis and chronic inflammation that maintains cellular and tissue function. Loss of parainflammation control contributes to dry AMD by invoking a chronic, heightened immune response that causes tissue destruction. The hallmarks of immune activation include drusen formation, subretinal and choroidal recruitment of microglia/macrophages, mast cell activation, and RPE immune activation[37–39]. This immune activation may involve close interplay between intracellular complement regulation and NLRP3 assembly in either immune cells or the RPE, although a recent study argues against NLRP3 inflammasome activation in the RPE[40]. At present, it is unclear at what point such immune activation converts from being protective to pathologic[41,42]. Figure 4 is a schema of the role of inflammation in AMD pathobiology.

**Complement system**. Genetic studies have identified complement pathway gene variants with AMD risk, which strongly implicates the complement pathway in driving AMD progression. However, the assumption that these variants induce excessive complement activation that leads to tissue injury remains unvalidated. For example, the exact mechanism for how the complement factor H (*CFH*) 402H variant contributes to dry AMD has eluded researchers for over a decade. *CFH* acts as both a cofactor during Factor I-mediated C3b cleavage and as a decay accelerating agent against the alternative pathway C3-convertase. The 402H variant increases complement activity[43]. However, it is unclear whether this increased activity induces elements of the dry AMD phenotype because *CFH* 402H also has impaired interaction with C-reactive protein, malondialdehyde, and heparan sulfate proteoglycans that can increase inflammation and lipoprotein accumulation[43–45]. As a mechanistic understanding of the causal molecular pathways is incomplete, perhaps not surprisingly, clinical trials testing complement inhibitors have failed[9].

**Mitochondrial dysfunction**. While age-associated mitochondrial dysfunction reduces cytoprotective pathway function, mitochondrial impairment severe enough to cause tissue damage will herald disease. In dry AMD, RPE mitochondrial mass is reduced, mitochondria are morphologically abnormal, and mitochondrial DNA (mtDNA) damage increases with disease severity[46,47]. In early and dry AMD, before a patient loses vision, mtDNA damage in genes involved in electron transport complex I, II, IV, and V in the RPE accumulates more in the macula than in the periphery, and occurs in the RPE prior to the neurosensory retina[47–49]. With mitochondrial loss and injury, the RPE has reduced baseline oxygen consumption and reserve capacity, and elevated ROS, making them poorly responsive to stress[50]. This change results in decreased ATP production, compromised calcium- and ROS-mediated

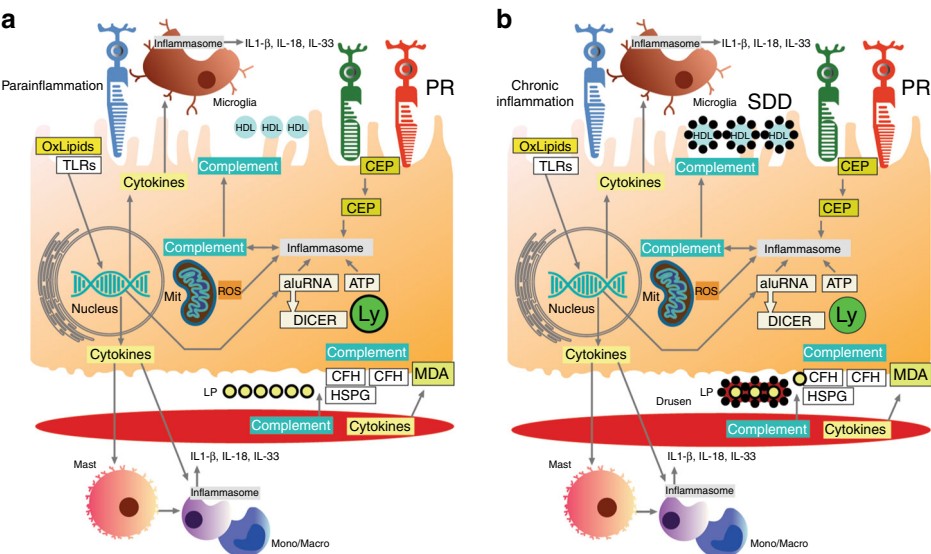

**Fig. 4** Cartoon of inflammatory processes involved in AMD pathobiology. In the subretinal space, oxidized lipids and excess HDLs can elicit an inflammatory response that includes microglial and systemic monocyte recruitment. **a** Initially, immune responses are heightened, representing parainflammation to maintain cell health. This includes activation of RPE, microglia, and potentially choroidal macrophages, with heightened intracellular cytokine responses (IL-18, IL-33), upregulation of autophagy, and immunometabolic regulation to maintain mitochondrial health. **b** Conversion from parainflammation to chronic inflammation can induce inflammasome activation by microglia, monocytes and the RPE (including IL-1b production) that contributes to subretinal drusenoid deposit formation. In the RPE, complement and the inflammasome can be activated by a number of triggers. In Bruch's membrane, the accumulation of lipoproteins and cellular debris elicits an inflammatory response from the RPE, mast cells, and monocytes/macrophages derived from the choroid or systemic circulation to activate both complement and the inflammasome. Inflammatory debris accumulates around lipoproteins/HAP particles during drusen formation in Bruch's membrane. CEP carboxyethyllysine, CFH complement factor H, HSPG heparan sulfate proteoglycan, Ly lysosome, MDA malondialdehyde, Mit mitochondria, mono/macro monocyte/macrophage, PR photoreceptor, ROS reactive oxygen species, SDD subretinal drusenoid deposit, TLR Toll-like receptor

signaling, and altered nucleotide metabolism, amino acid, lipid, and heme biosynthesis, which subsequently impair essential cytoprotective or specialized functions[51]. In addition, mitochondrial damage can activate apoptosis, leading to RPE degeneration and death, two key features of AMD.

**Cell death**. Photoreceptor cell death is the basis for permanent visual decline in dry AMD. Therefore, identifying the mechanisms involved in photoreceptor death is critical for developing new treatments to prevent permanent visual loss. Photoreceptor cell death is principally caused by apoptosis and necrosis[52]. Preventing photoreceptor cell death by specifically blocking apoptosis has been unsuccessful when inhibiting caspase because photoreceptor death is also caused by receptor-interacting protein kinase (RIPK)-mediated necrosis[53]. Likewise, when RIPK is inhibited, photoreceptor loss is unaffected[53]. In contrast, inhibiting both RIPK and caspases impairs both necrosis and apoptosis and rescues photoreceptors[53]. The mechanisms for RPE cell death may differ from those of photoreceptors. Double stranded RNA (dsRNA), a component of drusen, is a ligand for Toll-like Receptor-3, which mediates the innate immune response and cell death. After injecting dsRNA into the eye, photoreceptors die by apoptosis and RPE cells die by necrosis[54]. Since cell death pathways are redundant and complementary, combination therapies that block both apoptosis and necrosis may be effective for dry AMD.

**Strategies for understanding dry AMD etiology and pathogenesis**
**Genetics and genomics**. Genetic approaches have contributed enormously to our understanding of AMD pathobiology. The association of *APOE* variants with AMD was the first indication that a specific gene affected disease risk[55,56]. The advent of high-throughput technologies and agnostic genome-wide searches using family-based linkage and genome-wide association study (GWAS) approaches facilitated the identification of genes on chromosomes 1 (*CFH*)[57–60] and 10 (*ARMS2/HTRA1*)[61–64]. These discoveries are remarkable because they are among the few genetic associations for a common disease that increase the odds of disease by more than 2.5 among heterozygotes and 7.5 among homozygotes[60,62]. The association with *CFH* prompted focused investigation of complement system genes[27] which uncovered associations with *C3* [65,66], *BF/C2* [67], and *CFI* [68].

Current research has focused on understanding the role of specific genes and related pathways that drive AMD pathobiology[69–71]. The genes identified in a recent GWAS highlight the importance of both previously recognized and novel mechanistic pathways including complement activation, collagen synthesis, lipid metabolism/cholesterol transport, receptor-mediated endocytosis, and extracellular matrix organization[24]. Variants located in seven extracellular matrix genes were associated with advanced AMD and not early/intermediate AMD, suggesting that extracellular remodeling pathway(s) become relevant in advanced disease[72]. Pathway analysis and systems biology approaches have provided clues that upstream regulators/modulators of AMD risk variants are potential therapeutic targets[73,74]. These approaches give insight into the relationship between genetic variation and disease mechanism that can be tested in appropriate in vivo and in vitro models.

Except for *CFH* and *HTRA/ARMS2*, AMD-associated common variants confer a very small effect on disease risk (i.e., odds ratios <1.2), with as much as 70% of AMD risk remaining unexplained[24]. Increasing the samples for GWAS will undoubtedly implicate more loci in addition to the 40+ already identified[24,25,70,75–77], but the associated variants will account for an increasingly smaller amount of the genetic variance. A

portion of the proverbial "missing heritability" may be the result of gene interactions and environmental factors[78,79]. Associations with highly penetrant rare variants in genes known to harbor common variants have helped to establish causation, but the extent that rare genetic variants with high impact will account for the remaining genetic causality of AMD is unknown[24,80–84]. It will be difficult to identify rare, highly penetrant alleles in novel genes using traditional case–control designs because the requisite sample size to detect rare variants is much greater than aggregated cohorts assembled for GWAS. Nonetheless, such discoveries will provide additional insight into disease mechanisms.

Studies of non-European ancestry populations have demonstrated variability in the strength of genetic associations for AMD across Hispanics, Asians, and African Americans[77,85]. Protective alleles may contribute to these population differences. For example, AMD is very rare (<1%) in the isolated Timor-Leste population, and the few affected individuals do not harbor risk alleles at the two most associated AMD loci, CFH and ARMS2/HTRA1 (ref. [86]). However, the frequency of protective alleles at CFH, CFHR2, and C2 was increased in Timorese compared to outbred European ancestry populations, underscoring their potential importance. It would be valuable to study cohorts in which individuals with a high exposure to known AMD risk factors (e.g., smoking), advanced age, and elevated genetic risk for AMD do not develop AMD to gain insights into the protective mechanisms of specific genes or pathways that might serve as therapeutic targets.

Other types of genetic variants and mechanisms may have important roles in dry AMD. These include copy number variants or larger deletions and insertions, noncoding transcripts such as lncRNAs and microRNAs as modulators of gene expression and disease, and mitochondrial gene variants[21,87]. Somatic DNA modifications resulting from environmental exposures such as smoking and aging may also play a role in AMD pathogenesis[88,89]. Germline and somatic DNA changes may not directly impact pathogenic mechanisms, but rather influence expression or interact with other genes that are more proximate in pathways leading to disease. Thus, post-transcriptional DNA modifications that alter expression and their downstream consequences must be considered in studies that involve the transcriptome, post-translational modifications, proteome, and metabolome[90]. For example, using ATAC-Seq in the retina and RPE from AMD and control patients, Wang et al.[91] found global decreases in chromatin accessibility in the RPE from early AMD globes and in the retina of advanced disease, which suggests that RPE dysfunction drives disease onset. Cigarette smoke treatment of RPE cells recapitulated the chromatin accessibility changes seen in AMD, providing a potential epigenetic link of a known AMD risk factor with AMD pathology.

**In vitro dry AMD models.** The ability to differentiate embryonic stem (ES) and induced pluripotent stem (iPS) cells into RPE cells and retinal organoids has enhanced the ability to uncover the role of specific pathways and genetic influences. The use of patient-specific iPS cells is especially attractive because they have both the specific gene mutation of interest and the underlying genotype that could influence the mutation's contribution to the patient's disease and phenotype. Galloway et al. studied iPS-derived RPE cells from patients with and without AMD and found that sub-RPE basal deposits were more abundant and contained a lipid- and protein-rich "drusen-like" composition than unaffected cells[92]. Saini et al.[93] found that iPS-RPE cells derived from AMD patients with the ARMS2/HTRA1 AMD risk variant had increased complement and inflammatory proteins compared

donors without AMD. Saini et al.[94] also found a graded response to nicotinamide (NAM), with a marked inhibition of drusen-associated proteins, clusterin and vascular endothelial growth factor A, in iPS-RPE cells with the ARMS2/HTRA1 AMD risk variant. These experiments underscore the importance of targeting more than one disease pathway or mechanism at a time, while taking into consideration genotype risk[95]. Alternatively, the emergence of CRISPR/Cas9 gene editing has enabled researchers to further manipulate stem cell systems for mechanistic study. This approach can study the impact of a genetic variant by comparing its behavior with an isogenic control cell line without the mutation. With the possibility of personalized medicine, it may be possible to predict a drug response using iPS-derived cells or to differentiate iPS-derived retinal cells for transplantation into a diseased macula.

Improved culturing techniques have enabled the generation of retinal organoids. Using iPS cells, Zhong et al.[96] recapitulated spatiotemporally each of the main steps of retinal development observed in vivo during the formation of three-dimensional retinal cups that contain all major retinal cell types arranged in their proper layers. Remarkably, the photoreceptors showed outer segment formation and were photosensitive. This system can enable the study of cellular interactions.

The use of iPS cells also has limitations. Due to heterogeneity of iPS lines, a subclone could be selected that would misrepresent the cell's behavior. The epigenetic memory could remain and maintain elements of the cell's original phenotype. The relevance of iPS cells to an aging disease such as AMD is open to speculation since markers of cellular age, including mitochondrial function and telomere length, are reset to a youth-like state after old donor fibroblasts are reprogrammed to iPS cells[97].

**In vivo dry AMD models.** To provide mechanistic evidence of AMD pathobiology, zebrafish, mice, rats, and non-human primates are a few of the animals used to model AMD. The only animal to possess a macula, non-human primates are prohibitively expensive to maintain, and difficult to both manipulate and follow longitudinally since their life span is long, outliving the duration of most grants. Mice are the most practical organisms for modeling AMD due to their short life span, genetic manipulability, relatively low maintenance cost, and retinal architecture that is similar to humans. The mouse does not possess a macula, but rather a rod-rich retina that is similar to the rod-rich human parafovea. While the lack of a macula has been used to question their relevance to human AMD, mice have greatly aided our understanding of AMD pathobiology. Despite this important limitation, mice offer a valuable means of investigating the interplay of aging, oxidative stress, immunity, lipid metabolism, nutrition, and the microbiome as potential modifiers of retinal/RPE disease[43,98,99]. Such studies cannot be done with in vitro models and are not feasible in humans. However, most dry AMD mouse models represent early disease stages. Advanced age is the strongest risk factor for AMD and should be considered in any model. In fact, the most robust AMD models incorporate advanced age and are chronic in duration. Many AMD pathological hallmarks have been recapitulated in different mouse models, including vision loss; photoreceptor degeneration; RPE lipofuscin accumulation, hypo- and hyperpigmentation, multinucleation, and atrophy; choriocapillaris atrophy; and subretinal immune cell infiltration[43,100–103]. However, few models faithfully display a combination of these changes. While mice do not develop drusen, they do accumulate basal laminar deposits, which share many of the same constituents found in drusen[43,100,102]. The models that most closely simulate human AMD are based on known AMD-associated mechanisms, including inflammation,

lipid transport and metabolism, ECM remodeling, complement dysregulation, oxidative stress, and autophagy, which will enable study of how these factors interact with one another during lesion development, and how these lesions correlate with different stages of dry AMD[43,100,101]. However, because of differences in anatomy, metabolism, and the immune response between mice and humans, extrapolation to human disease should be made with caution.

**Clinical imaging of dry AMD**. Imaging is an essential component of the clinical assessment of AMD, and is used to diagnose, define disease severity and progression, assess treatment response, and complements mechanistic basic science analysis to understand AMD pathobiology. Currently, color fundus photographs are used to determine drusen size and total area, RPE pigmentary abnormalities, and GA, with the aim to categorize and predict AMD progression. Angiography with intravenously injected sodium fluorescein or indocyanine green (ICG) dyes provides detailed images of retinal and choroidal vasculature. Fundus autofluorescence identifies areas with excess fluorophores from RPE lipofuscin accumulation, loss of fluorophores from RPE loss, or fluorophore variation[104]. Reflectance-based structural optical coherence tomography (OCT), either spectral domain or the faster swept source OCT, provides micron-level resolution to enable detailed visualization of retinal layers, substructures, and AMD pathology[105]. OCT-angiography can provide detailed images of the choriocapillaris and choroidal abnormalities such as choriocapillaris dropout with GA[106]. By registering images, specific OCT features can be monitored in the same location over time to quantify morphologic progression[107].

Several new imaging modalities may provide novel insight into AMD pathobiology due to their ability to visualize key cellular and subcellular structures during different disease phases. Adaptive optics compensates for wave front aberrations to improve resolution[108]. When used with a scanning laser ophthalmoscope and OCT, adaptive optics systems can visualize individual cones, changes in rods, RPE, and SDD in dry AMD. Polarization-sensitive OCT highlights tissue using polarized signals and birefringence so that the RPE or fibrosis may be detected[109]. Fluorescence lifetime and hyperspectral fluorescence imaging, which measure dynamic metabolic states of the retina, offer new opportunities for retinal biomarkers that may complement other functional methods such as psychophysical testing (such as dark adaptation thresholds and kinetics) and electrophysiology[110]. These diagnostic tools may identify preclinical risk factors or subtypes of AMD that may differ with respect to progression and/or response to different treatments.

However, validating these new imaging technologies either individually or in combination has been restricted to small cohorts over short or inconsistent follow-up intervals[111,112]. Furthermore, imaging data have been simplified for clinical use, which limits their utility to uncover a deep understanding of AMD pathobiology. To advance our understanding, we must identify imaging biomarkers of early changes that reflect AMD pathobiology, predict disease progression and/or treatment response, and correlate with molecular markers that are relevant in both animal models and humans.

**Recommendations for an expedited, improved understanding, and clinical management of dry AMD**
Cellular and animal dry AMD models enable direct interrogation of pathological mechanisms in controlled environments, an attribute obviously lacking in human studies. The question remains of how to best leverage these models to determine the role of AMD-associated mechanisms that will lead to effective treatment. Cellular models are excellent for manipulating treatments that alter intracellular interactions while animal models enable the study of interactions among tissues, and how aberrant pathways induce an AMD phenotype. Since many models are incompletely or variably phenotyped, which can confuse interpretation, we advocate developing a consensus on the phenotypic standards of AMD animal models. This goal could be accomplished by a careful literature review on animal models that forms the foundation for roundtable discussions by leading scientists who utilize AMD animal models and clinicians who have expertise in AMD imaging, epidemiology, and treatment. Importantly, we recommend that animal and cell-based models rely on multiple risk factors to mimic the events at different stages of dry AMD, and that multiple models should be utilized to deepen the rigor beyond what a single model will provide.

**Systems biology approaches**. So far, genetic and epidemiologic studies have pinpointed more than 40 genetic variants and environmental and lifestyle factors, such as smoking, sunlight exposure, and a high-fat diet, that define individual risk for AMD. Integrating such heterogeneous factors that influence AMD pathobiology and turning this knowledge into prevention, prediction, and treatment is a big challenge. The next step is to understand the functional consequences of these risk factors. Traditionally, interlinks between genetic and non-genetic factors have been studied in a reductionist fashion. This approach is insufficiently comprehensive to assign the relative contribution of these risk factors to disease, the interaction of these factors, and the disease stage that these factors are pathogenic. In contrast, linking multiple data sources has already led to identifying the alternative complement pathway, ECM turnover, and lipid metabolism in AMD pathobiology[113]. The integration of genes carrying risk alleles into the framework of complex endophenotypes suggests that AMD risk genes act in multiple molecular pathways and large networks, associate with different anatomical microenvironments in the macula, and affect diverse higher order physiological activities such as Bruch's membrane homeostasis, protein and lipid turnover, energy metabolism, and complement regulation[114]. Thus, we recommend application of systems biology to address the complex interplay of pathogenic factors and provide the appropriate broad perspective needed to decipher the role of multiple factors in dry AMD pathogenesis. This approach would integrate basic, genomic, preclinical, medical, pharmacological, and clinical data into mathematical models of pathological processes at different stages of dry AMD in order to ask how relevant individual components act together within the living system. Research partnerships with companies that conduct dry AMD clinical trials could provide valuable phenotyping information that is required to assess disease progression and treatment response that could be integrated with genetics and AMD biomarkers. Furthermore, clinical trial participants should be encouraged to maintain clinical follow-up after the study's completion and to consider donating their eyes after death for research. While the effect of diet and environmental exposures on the human microbiome and the implications of these changes on the host immune and inflammatory systems have been explored in AMD[115], to gain a complete understanding of the complex interactions and their role in the pathogenesis and progression of dry AMD, all individual perturbations must be integrated into the entire system to establish a causal role.

Imaging data provide the foundation for a systems biology approach because they can identify the major pathological events in chronological sequence. These images should provide objective and reproducible longitudinal data that can be integrated with

large multisystems data from large cohorts. The imaging data must accurately co-localize morphologic changes in each of the multiple imaging modalities, and individual patient data must be cross-correlated before they can be incorporated into the overall analysis. Deep learning will expand the capability of analyzing these complex data sets[116]. To identify novel biomarkers, we recommend using agnostic, artificial intelligence methods that are independent of known biomarkers and/or prior categorizations to both confirm known mechanisms and identify novel associations. Because of the cost and scientific complexity of imaging modalities, we acknowledge that the analysis of large cohorts of imaging data may be limited to a few sites that are linked to a wide net of collaborating centers.

The imaging data must be integrated with content-rich clinical, epidemiological, pharmacological, and genetic data, as well as multi-omic approaches including genomic, transcriptomic, epigenomic, lipidomic, metabolomic, microbiome, and proteomic studies on fresh blood, donor eye tissue, or aqueous and vitreous biopsies from patients with dry AMD. Advances in multi-scale data integration of large throughput experiments and analysis have opened new avenues for discovery and have bridged genetic risk variants with expression and protein level information[117,118]. We recommend leveraging large data sets, whether public or proprietary, that are now available including those containing common and rare genetic variants for AMD, transcriptomic data derived from the mouse and eventually human retina, and proteomic data for developing a systems biology model of dry AMD[119]. The emergence of single-cell RNA and ATAC sequencing offers the possibility of defining the relative contributions of individual cell types to AMD pathogenesis, and will enrich the models that are developed. Finally, pathogenic molecular pathways that have been interrogated in cell and animal models must be integrated with these data, which must be quality controlled and consistently curated. These efforts require centralized and shared resources to disseminate the information, and could provide an unprecedented resource to understand AMD pathobiology, re-define stratified clinical phenotypes that are based on molecular and dynamic parameters, develop combinatorial biomarkers or biomarker signatures, and initiate novel diagnostic, prevention, prediction, and therapeutic approaches where reductionist approaches have failed.

**The need for a personalized approach to treat dry AMD.** Since dry AMD is a multifactorial disease resulting from perturbations in multiple pathways, the pathogenic signals are likely to vary among individuals and at different disease stages. Thus, we advocate designing computer models that are tailored to the individual so that physicians can accurately predict dry AMD risk, disease progression, and response to treatment. We believe that this precision medicine approach for AMD, which was unimaginable just a few years ago, is realistic and similar to what is being achieved in other complex diseases such as cancer[120] or diabetes[121]. These individualized risk prediction models may potentially enable targeting patients with specific dry AMD subtypes for tailor-made therapies. Genomic and proteomic biomarkers, and other risk factors identified from a systems approach could monitor the effectiveness of preventive therapies to modify dry AMD risk before the onset of vision-threatening complications. Once single markers or marker signatures are linked to specific stages and molecular mechanisms of AMD, they can be employed as predictive or companion biomarkers and biomarker signatures that hallmark specific pathogenic features or pathological mechanisms of AMD. Having a set of multimodal markers at hand to group and stratify patients according to their

individual risk and expected response pattern, therapeutic trials tailored to specific patient groups could be designed. Individuals within high-risk groups could be better motivated to comply with an early intervention on the basis of a predicted cumulative risk and a likely positive expected response to therapy prior to losing vision. The prerequisite for this rationale, however, depends upon the successful advancement of promising, preclinical drugs into clinical trials.

**Benefits and challenges of a systems biology approach for dry AMD.** This approach would be a departure from the current research paradigm. Such an approach would require the integrated collaboration of leading clinicians, imaging experts, a wide variety of basic scientists, bioinformaticians, and biostatisticians among other necessary expertise. Besides vision scientists, this research will benefit by recruiting scientists outside of dry AMD research who will provide not only additional technical expertise, but potentially a "Medici effect" (i.e., the most important innovations occur when concepts from diverse disciplines are conjoined) where scientists without AMD expertise could provide disruptive innovation that would complement the conventional approach. Due to the perceived cost, the clinical and basic science may be conducted at only a few centers that have the appropriate infrastructure to implement this large-scale approach, or require coordinated efforts of investigators at multiple institutions. However, implementing large-scale team systems biology research should be possible. We can now link heterologous large data sets to acquire novel information on which to generate new hypotheses, calculate risks on an individual basis, identify drivers of disease manifestation and progression, and turn knowledge into risk assessment and clinical recommendations. The EYE-RISK consortium is an example of how this goal can be achieved. A consortium of participants from different European countries, EYE-RISK (www.eye-risk.eu) funded by the EU Horizon 2020 program, uses a multidisciplinary approach to examine comprehensive epidemiologic data and biosamples from large European epidemiologic eye cohorts and biobanks (total study population $N = 100,000$). Collaborative studies conducted by EYE-RISK, the International AMD Genomics Consortium, and the Three Continent Consortium have been instrumental in identifying genetic risk variants and assessing environmental risk factors.

Currently, there is no organized, cost-effective system for obtaining human donor globes that have short death-to-dissection times with high-quality RNA, metabolite, DNA, and protein that is adequate to meet the needs of the AMD research community[122]. Likewise, the few existing repositories of blood or ocular fluid samples from well-phenotyped patients for study are not sufficiently resourced to distribute specimens on a wide scale. To enable this systems approach, we recommend developing mechanisms to create these libraries of AMD eyes, blood and ocular fluid samples, an array of omics data (e.g., genomic, transcriptomic, methylomic, proteomic, etc.) derived from these samples, and a program that ensures their effective use for dry AMD research. The National Cell Repository for Alzheimer disease which banks blood and DNA specimens, brain tissue, and associated data for Alzheimer disease research (https://www.nia.nih.gov/research/resource/national-cell-repository-alzheimers-disease-ncrad) is one example of a coordinated approach to address this need.

The current National Eye Institute and foundation-based funding mechanisms for AMD research are largely confined to grants awarded to individual researchers for a defined amount of money. This approach, which has been the cornerstone of our

current understanding of AMD pathobiology to date, enables researchers to address specific, focused mechanistic questions, but in relative isolation. A systems biology approach using sophisticated high-throughput "-omics" assays is not feasible with a typical individual R01 NIH grant. We recommend that the NIH and other funding agencies examine how they allocate precious resources and consider including large programs that would enable the integration of basic discovery research, mechanistic investigation using well-phenotyped human populations, and suitable animal and cell-based models that are needed to gain a detailed understanding of dry AMD pathobiology. Mechanisms that foster access and coordination of multiple investigators to resources that would not be achievable on an individual basis would address this next step in dry AMD research. The implementation of this approach could be a template for tackling other complex diseases.

## Conclusion

Future directions in dry AMD research should emphasize systems biology approaches that integrate omic, pharmacological, and clinical data into mathematical models that can predict disease onset and progression, identify biomarkers, establish disease causing mechanisms, and monitor response to therapy. Success in these areas will likely be achieved most expediently and effectively by promoting collaborative efforts of multidisciplinary investigator teams and developing centralized resources including clinical, imaging, omic, and other types of data as well as carefully phenotyped eye tissue from large cohorts of patients with and without AMD.

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

## Acknowledgements

We thank Paul Sieving, MD, Ph.D., Director of the NEI and the National Eye Institute for organizing the AMD Pathobiology group. We also thank Anna E. Mazzucco, Ph.D. for her contributions to the group. J.T.H.: NIH EY027691; NIH R42 EY029625-01; BrightFocus Foundation; Macular Degeneration Foundation; Wilmer-Bayer Alliance Grant, Bayer Pharmaceuticals, Inc.; Unrestricted Grant from Research to Prevent Blindness (Wilmer Eye Institute), Robert Bond Welch Professorship. C.B.R.: NIH R01 EY026161; P30 EY005722 to Duke University, a Research to Prevent Blindness (RPB)/ International Retinal Research Foundation (IRRF) Catalyst Award for Innovative Research Approaches for AMD, an unrestricted grant from RPB (to the Duke Eye Center), and a Fighting Blindness Individual Investigator Award. A.D.D.: NIHR Biomedical Research Centre Moorfields Eye Hospital and UCL-Institute of Ophthalmology, National Eye Research Centre UK, Macular Society UK, Medical Research Council UK, Rosetrees Trust UK. M.B.G.: Harold and Pauline Price Foundation, Research to Prevent Blindness, NY, NY, NIH/NEI R01 EY09859 Gorin (PI). J.W.M.: NEI Core Facility Grant EY014104, Yeatts Retina Fund, Research to Prevent Blindness, Retina Research Fund, Champalimaud Vision Award. C.A.T.: an unrestricted grant from RPB (to the Duke Eye Center). M.U.: EYE-RISK is funded by the Horizon 2020 program of the European Union. Funding is provided to Marius Ueffing in the period of 2015–2019 under Grant Agreement number 634479. M.Z.: Joseph J. and Marguerite DiSepio Retina Research Fund; Eng Family Foundation; New Jersey Lions Eye Research Foundation; Paid Consultant for: California Institute of Regenerative Medicine, Cell Cure, Chengdu Kanghong Biotechnology Co., Coherus Biosciences, Inc., Daiichi Sankyo, Frequency Therapeutics, Foundation Fighting Blindness, Genentech/Roche, Healios KK, Inc., Iridex, Isarna Therapeutics, Makindus, Novartis Pharma AG, Ophthotech Corp., Percept Corp. L.A.F.: NIH U01-AG032984, NIH UF1-AG046198, NIH R01-AG048927, NIH RF1-AG057519.

## Author contributions

All authors made substantial contributions to the conception and design of the work, and have contributed to the manuscript writing and revisions.

## Additional information

**Competing interests:** J.T.H.: Grant funding from Bayer Pharmaceuticals, Inc. J.W.M.: Consultant/Advisor for Genentech/Roche, Bausch+Lomb, Kalvista Pharmaceuticals, ONL Therapeutics; Grant Support from Lowy Medical Research Institute; Equity in ONL Therapeutics; Patents/Royalties from ONL Therapeutics/Mass. Eye and Ear, Valeant Pharmaceuticals/Mass. Eye and Ear. C.A.T.: Alcon royalties for surgical technologies. M. Z.: Cell Cure, Chengdu Kanghong Biotech, Coherus Biosciences, Daiichi Sankyo, Frequency Therapeutics, Healios KK, Iridex, Isarna Therapeutics, Genentech/Roche, Makindus, Novartis Pharma AG, Ophthotech, Percept Corp., Rutgers University (patent). The remaining authors declare no competing interests.

