## [Peer Review File · Nature Communications]

Reviewers' Comments:

Reviewer #1:

Remarks to the Author:

The authors should be congratulated on a very thorough and detailed discussion of non-exudative AMD, presented in an extremely comprehensive clear style, and superbly referenced. The perspective of the authors is clearly conveyed and appropriately so. However, I would encourage the authors to consider the following suggestions for the addition and/or revision of some aspects of the manuscript.

1. AMD is clearly a disease influenced by the effects of multiple genes, but it is no more a genetic disease than many other polygenic disorders, such as osteoarthritis. It is a disease of AGING in which we have insights into many mechanisms of the pathobiology of the disease but no understanding of underlying etiology. Thus, GWAS screening of large populations is unlikely to identify its etiology but rather, as the authors state, identify other variant genes that are associated with a minor effect on incidence. The mechanistic complexity of AMD suggests that therapeutic intervention without understanding etiology may not be fruitful.
2. The emphasis on system biology to explore the etiology of the disease is most appropriate, and in vitro study of single cells (iPS, ESC or terminally differentiated) or retinal organoids is most unlikely to be insightful. As the authors correctly emphasized there are so many in vivo host mechanisms involved in retinal cell protection or death that studying pathobiology in their absence is not likely to be fruitful.
3. The research study of AMD has been most likely hindered by the dominant paradigm of the day - namely that it is a disease of the RPE and not the neurosensory retina, specifically the photoreceptors. The authors correctly draw attention to the parafoveal rod photoreceptors as the first cells to die in AMD, and accept the common popular hypothesis that RPE dysfunction is the cause of the disease. It is certainly understandable since the first and most obvious clinical signs of AMD are in the RPE and Bruch's membrane. However, it may be that the early documented parafoveal mitochondrial abnormalities in RPE may be the consequence of rod photoreceptor dysfunction, and not the cause of rod photoreceptor demise, as perhaps are many of the other RPE abnormalities characteristic of AMD. As has been recently demonstrated in Retinitis Pigmentosa abnormalities in rod photoreceptors can result in metabolomic abnormalities in RPE that ultimately result in cone dormancy and/or death. A major paradigm shift in etiology may be necessary to unravel the cause of AMD, and thus, this issue might be appropriate to address.
4. The recommendation of a consortium of scientists, clinicians, epidemiologists, etc. working together to advance our understanding is most reasonable. Indeed, it may be appropriate to recommend a "War on AMD" where the NIH would fund such a consortium under the leadership of a Director, including recognized experts and their laboratories in many of the disciplines appropriate to study this disease.

I have not commented on specific sentences or paragraphs in the manuscript because I think that it is very well written.

Henry J Kaplan, MD

Reviewer #2:

Remarks to the Author:

This perspective is a white paper based on a National Eye Council group to assess current and future research directions for dry (geographic) atrophy; one of the main end stages of the eye disease age related macular degeneration (AMD). There are currently no effective treatments for dry AMD and hence the emphasis on this subtype.

The manuscript on the whole is well written and clear in presentation although it could be aided by the inclusion of flow diagrams or diagrams to indicate the complexity of each of the aspects that are presented. In particular in the section "the role of lipids" and in the "immune response parts"

as these would facilitate easier reading.

The majority of the manuscript details current knowledge of the field broken into a number of subject areas which cover: pathological changes, imaging, oxidative stress, lipids, inflammation, mitochondrial dysfunction, cell death, genetics, in-vitro and in-vivo models. The last few pages are focussed on recommendations through a systems biology approach, biomarker identification and personalized medicine.

While it was a straightforward read, I didn't really find any fundamental new knowledge that was not already in the literature. There were also some areas that were overlooked such as epigenetics in AMD which have recently been published in Nature Comm and other potential mechanisms of phagocytosis and scavenger effect such as P2X7 or on caveats to many of the approaches indicated such as effective ageing of iPSCs.

Other points

Line 59-61, sentence doesn't make sense, needs rewriting

Line 73. What do you mean by "time" and "multiple medium"?

Line 80, some references needed for the negative phase 3 studies

Line 94, "to" missing before 'report'

Line 126, why are the papers cited 'recent' when they range in date from 1995 to 2012?

Can you provide a figure to detail the events of pathological change for clarity.

Line 202, the genes described for oxidative stress were not identified in the IAMDGC study (Fritsche et al, 2016) – what does this mean in terms of their role in disease?

Lines 260-262 appear to repeat lines 243-244

Line 332, a word missing after 'future research...'

Line 340, this statement not supported by the previous statement about NLRP3 on line 289

Lines 336-346, appears to be a series of sentences with limited connection – need to rewrite

Line 434, indicate what 7 extracellular matrix genes

Line 444, the number of genes is indicated as 20 then on line 576 it is 40 genes but neither of these tie in with the number of genetic variants described by the IAMDGC.

Line 466-469, if there are elevated genetic risk variants then why are they gaining insights into protective mechanisms – sentence seems to be contradicting itself

Line 520, include some references

What are the disadvantages of using iPSCs?

Reviewers' comments:

Reviewer #1 (Remarks to the Author):

The authors should be congratulated on a very thorough and detailed discussion of non-exudative AMD, presented in an extremely comprehensive clear style, and superbly referenced. The perspective of the authors is clearly conveyed and appropriately so. However, I would encourage the authors to consider the following suggestions for the addition and/or revision of some aspects of the manuscript.

We thank the reviewer for the favorable impression.

1. AMD is clearly a disease influenced by the effects of multiple genes, but it is no more a genetic disease than many other polygenic disorders, such as osteoarthritis. It is a disease of AGING in which we have insights into many mechanisms of the pathobiology of the disease but no understanding of underlying etiology. Thus, GWAS screening of large populations is unlikely to identify its etiology but rather, as the authors state, identify other variant genes that are associated with a minor effect on incidence. The mechanistic complexity of AMD suggests that therapeutic intervention without understanding etiology may not be fruitful.

We agree.

2. The emphasis on system biology to explore the etiology of the disease is most appropriate, and in vitro study of single cells (iPS, ESC or terminally differentiated) or retinal organoids is most unlikely to be insightful. As the authors correctly emphasized there are so many in vivo host mechanisms involved in retinal cell protection or death that studying pathobiology in their absence is not likely to be fruitful.

We agree.

3. The research study of AMD has been most likely hindered by the dominant paradigm of the day - namely that it is a disease of the RPE and not the neurosensory retina, specifically the photoreceptors. The authors correctly draw attention to the parafoveal rod photoreceptors as the first cells to die in AMD, and accept the common popular hypothesis that RPE dysfunction is the cause of the disease. It is certainly understandable since the first and most obvious clinical signs of AMD are in the RPE and Bruch's membrane. However, it may be that the early documented parafoveal mitochondrial abnormalities in RPE may be the consequence of rod photoreceptor dysfunction, and not the cause of rod photoreceptor demise, as perhaps are many of the other RPE abnormalities characteristic of AMD. As has been recently demonstrated in Retinitis Pigmentosa abnormalities in rod photoreceptors can result in metabolomic abnormalities in RPE that ultimately result in cone dormancy and/or death. A major paradigm shift in etiology may be necessary to unravel the cause of AMD, and thus, this issue might be appropriate to address.

We agree that this possibility exists.

4. The recommendation of a consortium of scientists, clinicians, epidemiologists, etc. working together to advance our understanding is most reasonable. Indeed, it may be appropriate to recommend a "War on AMD" where the NIH would fund such a consortium under the leadership of a Director, including recognized experts and their laboratories in many of the disciplines appropriate to study this disease. I have not commented on specific sentences or paragraphs in the manuscript because I think that it is very well written.

Henry J Kaplan, MD

We agree with Dr. Kaplan that a mechanism which enables multiple investigators to study AMD on a large scale.

Reviewer #2 (Remarks to the Author):

This perspective is a white paper based on a National Eye Council group to assess current and future research directions for dry (geographic) atrophy; one of the main end stages of the eye disease age related macular degeneration (AMD). There are currently no effective treatments for dry AMD and hence the emphasis on this subtype.

The manuscript on the whole is well written and clear in presentation although it could be aided by the inclusion of flow diagrams or diagrams to indicate the complexity of each of the aspects that are presented. In particular in the section "the role of lipids" and in the "immune response parts" as these would facilitate easier reading.

We appreciate the favorable impression to our work. We have added two figures (Fig 3 and 4) to guide the readers in the “lipids” and “immune response” sections.

The majority of the manuscript details current knowledge of the field broken into a number of subject areas which cover: pathological changes, imaging, oxidative stress, lipids, inflammation, mitochondrial dysfunction, cell death, genetics, in-vitro and in-vivo models. The last few pages are focussed on recommendations through a systems biology approach, biomarker identification and personalized medicine.

While it was a straightforward read, I didn't really find any fundamental new knowledge that was not already in the literature. There were also some areas that were overlooked such as epigenetics in AMD which have recently been published in Nature Comm and other potential mechanisms of phagocytosis and scavenger effect such as P2X7 or on caveats to many of the approaches indicated such as effective ageing of iPSCs.

To provide novelty in reporting of our collective knowledge of AMD, as suggested, we report the following new findings:

1. The recent finding of reduced chromatin unwinding in promoters of the RPE that correlate with the RPE transcriptome from AMD donor eyes.
2. P2X7 receptor as a potential target that can influence both lysosomal function and inflammasome activation.
3. Techniques for aging iPS cells.
 - recent studies have presented evidence that markers of cellular age, including mitochondrial fitness and telomere length, are reset to a young-like state when old donor fibroblasts are reprogrammed to iPSCs as reviewed in¹
 - Miller and Studor showed that overexpression of progerin in iPS fibroblasts induces an aged-like state²

Other points

Line 59-61, sentence doesn't make sense, needs rewriting.

We have modified the sentence to now read:

While AMD is **currently** more prevalent in Europe and North America than Asia, because Asia accounts for more than 60% of the world population, the largest projected number of AMD cases will occur in Asia

Line 73. What do you mean by “time” and “multiple medium”?

We have modified as follows:

The presence of medium-sized drusen are risk factors for developing large drusen, and the presence of multiple medium- or large-sized drusen, and RPE pigmentary changes are independent risk factors for developing late AMD³. In addition, the duration of AMD, whether with medium or large drusen, is an additional risk factor for progressing to late AMD.³

Line 80, some references needed for the negative phase 3 studies

We are unaware of any publications reporting the negative results of the Phase 3 Lampalizumab trial. However, we have added description of other failed clinical trials in complement and other targets.

Line 94, "to" missing before 'report'

This sentence has been omitted due to the editor's recommendations.

Line 126, why are the papers cited 'recent' when they range in date from 1995 to 2012?

We have removed "recent".

Can you provide a figure to detail the events of pathological change for clarity.

We have included additional figures of hard drusen, soft druse, and SDD into a revised Fig 2.

Line 202, the genes described for oxidative stress were not identified in the IAMDGC study (Fritsche et al, 2016) – what does this mean in terms of their role in disease?

Not all genes with a role in AMD have been or will be in the future identified by GWAS because disease-associated variants in these genes are rare (and thus current GWAS samples have low power to detect them), effect AMD risk by interaction with other genes, or as is the case for some oxidative stress genes that are mitochondrial are not typically evaluated in GWAS. As mentioned in the sentence, we believe that these genes contribute only a small proportion of attributable risk.

Lines 260-262 appear to repeat lines 243-244

The original sentence in lines 243-244 was modified slightly by the recommendations of the editors.

Line 332, a word missing after 'future research...'

We have added "should" to the sentence.

Line 340, this statement not supported by the previous statement about NLRP3 on line 289

We respectfully disagree. In original line 289, we point out one study that raises question whether NLRP3 can be identified in RPE cells. However, the beginning of this sentence does provide evidence for its existence in RPE cells.

Lines 336-346, appears to be a series of sentences with limited connection – need to rewrite

This has been rewritten on the advice of the editors

Line 434, indicate what 7 extracellular matrix genes

Since we have been asked to reduce the “review” aspects of this manuscript, we have not included these genes.

Line 444, the number of genes is indicated as 20 then on line 576 it is 40 genes but neither of these tie in with the number of genetic variants described by the IAMDGC.

The correct number of genes identified by GWAS is 40+. This includes 34 reported by the IAMDGC and other loci discovered in other GWAS. This number does not include several other genes identified through candidate gene analysis and other approaches. We changed the number of genes on line 444 from 20 to 40 and added several references to reinforce that not all significantly associated AMD loci identified by GWAS were reported by the IAMDGC.

Line 466-469, if there are elevated genetic risk variants then why are they gaining insights into protective mechanisms – sentence seems to be contradicting itself

The sentence does not indicate that the high risk variants provide insight into protective mechanisms, but rather highlights the observation stated in the previous sentence that there are populations such as the Timorese that don't develop AMD despite high exposure to known AMD risk factors. The Timorese lack risk alleles at the two most potent loci (*CFH* and *ARMS2/HTRA1*) and have known protective alleles at several AMD loci. Studies of the Timorese and other populations with low frequency of AMD may provide insight into protective mechanisms.

Line 520, include some references

We are not aware of specific references at this point, but it is a scientifically logical approach.

What are the disadvantages of using iPSCs?

We have added the following paragraph to address disadvantages of iPSCs.

However, the use of iPS cells has some limitations. Current culture techniques are expensive and time-consuming. It is possible that with the heterogeneity of iPS lines, a subclone could be selected that would provide

misleading results. Reprogrammed cells might maintain epigenetic memory that will influence the cell's original phenotype.⁴ Finally, the relevance of iPS cells to an aging disease such as AMD is open to speculation. Recent studies have found that markers of cellular age, including mitochondrial function and telomere length, are reset to a youth-like state when old donor fibroblasts are reprogrammed to iPS cells.¹ However, Miller and Studer showed that overexpression of progerin in iPS fibroblasts induces an aged-like state.² Whether the aging phenotype is recapitulated is unknown.

Sincerely,

James T. Handa and Lindsay Ferrar

REFERENCES

1. Mahmoudi, S. & Brunet, A. Aging and reprogramming: a two-way street. *Curr Opin Cell Biol* **24**, 744-756 (2012).
2. Miller, J. & Studer, L. Aging in iPS cells. *Aging (Albany NY)* **6**, 246-247 (2014).
3. Chew, E.Y., *et al.* Ten-year follow-up of age-related macular degeneration in the age-related eye disease study: AREDS report no. 36. *JAMA Ophthalmol* **132**, 272-277 (2014).
4. Tan, L., *et al.* Naked Mole Rat Cells Have a Stable Epigenome that Resists iPSC Reprogramming. *Stem Cell Reports* **9**, 1721-1734 (2017).

Reviewers' Comments:

Reviewer #2:

Remarks to the Author:

The review is greatly improved but a few points remain.

It would be useful to add a few lines on the limitation of current approaches early in the piece and emphasise why these will not be appropriate going forward

Need a section of serum/biomarkers and proteomics

What recommendations were fed back to the National Eye Advisory Council – please include in dot point

Section beginning line 753 “Developing resources for Clinical Research” would seem limited in that it concentrates on EYE-RISK. What other eye consortia are there and what have they tried to achieve/achieved?

Line 104. Please provide more details about the National Advisory Eye Council eg who it represents, the date of the meeting, the specific objectives of this meeting for GA

Line 343. Please relate the findings on apoptotic death to changes in figure 3

Line 262. CHF should read CFH

Line 371. Rewrite end of sentence as “more effective treatment paradigms for dry AMD”

Line. Include epigenetics in what might be missing

Line 412. Several studies have already reported rare variant associations in AMD and so these and how they relate to family studies should be mentioned

Line 766. If EYE-RISK, the AMD Consortium and the Three Continents studies have made achievements then is there a paper to document this collaboration? Also if there are 4 aims for EYE-RISK but funding runs out this year then it would make sense to document in brief what has been achieved

Line 792. Indicate that this could take a very long time for a person to be able to actually donate their eyes and hence what other strategies might be put in place to mitigate this timing.

Reviewer #2:

The review is greatly improved but a few points remain.

- *We thank the reviewer.*

It would be useful to add a few lines on the limitation of current approaches early in the piece and emphasise why these will not be appropriate going forward

- *We have made statements describing the limitations of current approaches at the section (p.) recommended by the editors. We respectfully disagree that these reductionist approaches are inappropriate, but instead, will have a defined role in validating mechanism of findings that results from a broader systems biology approach.*

Need a section of serum/biomarkers and proteomics

- *We certainly agree with the importance of serum/biomarkers and proteomics in our quest to understand AMD pathobiology. In fact, the advantages of these concepts and their importance in our recommended approach of using a systems biology approach, appeared in the original version, and due to its importance, they remain in this revised version. However, due to space constraints and the significant reduction in manuscript length, we were unable to include a separate section.*

What recommendations were fed back to the National Eye Advisory Council – please include in dot point

- *A summary of our discussions has been presented to the NEAC at three council meetings. We have included a summary of our specific recommendations in a Box.*

Section beginning line 753 “Developing resources for Clinical Research” would seem limited in that it concentrates on EYE-RISK. What other eye consortia are there and what have they tried to achieve/achieved?

- *We focused on EYE-RISK because it is the largest consortium that has focused on AMD, and we described it to illustrate a potential platform that could be utilized. While we could have kept a short description of Macustar, another eye consortium, due to space constraints, we have removed this section. The EYE-RISK information remains in the section “Benefits and challenges of a systems biology approach for dry AMD” on the recommendation of the editors.*

Line 104. Please provide more details about the National Advisory Eye Council eg who it represents, the date of the meeting, the specific objectives of this meeting for GA

- *As mentioned on the National Eye Institute website, The National Advisory Eye Council “advises, assists, consults with, and makes recommendations to the Secretary of Health and Human Services (Secretary) and the Director, National Eye Institute on matters related to the activities carried out by and through the Institute and the policies respecting*

these activities". The Council meets 3x per year. The NAEC has not specifically had a meeting for GA as the reviewer suggests, but rather, gave the NEI director permission to put the AMD Pathobiology group together. The group has met regularly over 2 years, which enabled the perspective that has been written, and has designs on meeting in the future so help develop a roadmap to implement these ideas.

We hope the reviewer has a better understanding of the NAEC's role. However, we have not included this information since it does not fit with the intent of our perspective.

Line 343. Please relate the findings on apoptotic death to changes in figure 3

- The intent of figure 3 is to summarize the role of lipids in the biogenesis of drusen and reticular pseudodrusen. The link between these lipid pathways has not been conclusively linked to apoptotic cell death, so we have not included it in our figure.

Line 262. CHF should read CFH

The sentence containing this typo has been removed.

Line 371. Rewrite end of sentence as "more effective treatment paradigms for dry AMD"

- This sentence has been edited out of the revised version.

Line. Include epigenetics in what might be missing

- While the specific line was not provided, we believe that we included "epigenetics" in the section on reduced chromatin unwinding in AMD that this reviewer suggested including in the first review. This section appears on page 13.

Line 412. Several studies have already reported rare variant associations in AMD and so these and how they relate to family studies should be mentioned

- We would like to include this information, but we needed to significantly reduce the section on genetics as well as meaningful references. That said, most of these findings were observed in genes that were previously implicated in AMD by studies of common variants (e.g., complement pathway genes, COL8A1). We discuss the role of rare variants in AMD on p. 12.

Line 766. If EYE-RISK, the AMD Consortium and the Three Continents studies have made achievements then is there a paper to document this collaboration? Also if there are 4 aims for EYE-RISK but funding runs out this year then it would make sense to document in brief what has been achieved

- The EYE-RISK has made achievements that have been summarized in the following publications.

*Brown CN, Green BD, Thompson RB, den Hollander AI, Lengyel I; **EYE-RISK** consortium.*

Metabolomics and Age-Related Macular Degeneration. Metabolites. 2018 Dec 27;9(1). pii: E4. doi: 10.3390/metabo9010004.

*Colijn JM, den Hollander AI, Demirkan A, Cougnard-Grégoire A, Verzijden T, Kersten E, Meester-Smoor MA, Merle BMJ, Papageorgiou G, Ahmad S, Mulder MT, Costa MA, Benlian P, Bertelsen G, Bron AM, Claes B, Creuzot-Garcher C, Erke MG, Fauser S, Foster PJ, Hammond CJ, Hense HW, Hoyng CB, Khawaja AP, Korobelnik JF, Piermarocchi S, Segato T, Silva R, Souied EH, Williams KM, van Duijn CM, Delcourt C, Klaver CCW; European Eye Epidemiology Consortium; **EYE-RISK** Consortium. Increased High-Density Lipoprotein Levels Associated with Age-Related Macular Degeneration: Evidence from the EYE-RISK and European Eye Epidemiology Consortia. *Ophthalmology. 2019 Mar;126(3):393-406.**

*Merle BMJ, Colijn JM, Cougnard-Grégoire A, de Koning-Backus APM, Delyfer MN, Kieft-de Jong JC, Meester-Smoor M, Féart C, Verzijden T, Samieri C, Franco OH, Korobelnik JF, Klaver CCW, Delcourt C; **EYE-RISK** Consortium. Mediterranean Diet and Incidence of Advanced Age-Related Macular Degeneration: The EYE-RISK Consortium. *Ophthalmology. 2019 Mar;126(3):381-390.**

*Colijn JM, Buitendijk GHS, Prokofyeva E, Alves D, Cachulo ML, Khawaja AP, Cougnard-Grégoire A, Merle BMJ, Korb C, Erke MG, Bron A, Anastasopoulos E, Meester-Smoor MA, Segato T, Piermarocchi S, de Jong PTVM, Vingerling JR, Topouzis F, Creuzot-Garcher C, Bertelsen G, Pfeiffer N, Fletcher AE, Foster PJ, Silva R, Korobelnik JF, Delcourt C, Klaver CCW; **EYE-RISK** consortium; European Eye Epidemiology (E3) consortium. Prevalence of Age-Related Macular Degeneration in Europe: The Past and the Future. *Ophthalmology. 2017 Dec;124(12):1753-1763.**

Regarding our perspective, we have significantly reduced this section and have not provided some of the original details since the original intent is to provide a sample of a direction that could be implemented to utilize systems biology. Thus, we respectfully suggest that the requested information needed to be omitted.

Line 792. Indicate that this could take a very long time for a person to be able to actually donate their eyes and hence what other strategies might be put in place to mitigate this timing.

- The reviewer makes a compelling point regarding the long interview of receiving the globe of a patient who participated in a clinical trial. We believe that this is only one part of the strategy that we have outlined so that delays in getting the tissue will not be a major impediment. Regarding donor globes, on page 24, we have explained in detail the need for well-characterized clinical histories with multi-modal imaging information of high quality donor globes.

We believe the revised manuscript is more focused and has appropriate emphasis on future directions with our specific recommendations. We hope that the manuscript is now suitable for publication.

Reviewers' Comments:

Reviewer #2:

Remarks to the Author:

The authors have now completed all the relevant points.

Minor point, line 549 "which has" was repeated twice and should be removed

Francesco Conti, PhD
Associate Editor
Nature Communications

June 15, 2019

Dear Dr. Conti,

We thank the editors again for the opportunity to revise our manuscript. Below is our response to reviewer 2.

Reviewer #2:

The authors have now completed all the relevant points.

We thank the reviewer for the helpful guidance.

Minor point, line 549 "which has" was repeated twice and should be removed

We have addressed this.

We hope that the manuscript is now suitable for publication.

Sincerely,

James T. Handa and Lindsay Farrer